# InstructAV: Instruction Fine-tuning Large Language Models for Authorship Verification

**Yujia Hu** [*], **Zhiqiang Hu** [*], **Chun-Wei Seah, Roy Ka-Wei Lee**
Information Systems Technology and Design
Singapore University of Technology and Design
{yujia_hu, chunwei_seah, roy_lee}@sutd.edu.sg
{zhiqiang_hu}@mymail.sutd.edu.sg

## Abstract

Large Language Models (LLMs) have demonstrated remarkable proficiency in a wide range of NLP tasks. However, when it comes to authorship verification (AV) tasks, which involve determining whether two given texts share the same authorship, even advanced models like ChatGPT exhibit notable limitations. This paper introduces a novel approach, termed InstructAV, for authorship verification. This approach utilizes LLMs in conjunction with a parameter-efficient fine-tuning (PEFT) method to simultaneously improve accuracy and explainability. The distinctiveness of InstructAV lies in its ability to align classification decisions with transparent and understandable explanations, representing a significant progression in the field of authorship verification. Through comprehensive experiments conducted across various datasets, InstructAV demonstrates its state-of-the-art performance on the AV task, offering high classification accuracy coupled with enhanced explanation reliability.

## 1 Introduction

Authorship Verification (AV) is a task aimed at determining if two texts were written by the same author, with significant implications across forensics, literature, and digital security domains (Halvani et al., 2019; Stamatatos, 2016). Traditionally, AV relied on stylometric analysis, utilizing linguistic and stylistic features, such as word and sentence lengths, and function word frequencies, to distinguish between authors (Seroussi et al., 2011; Bevendorff et al., 2019). However, the advent of machine learning, particularly deep learning models like BERT (Devlin et al., 2018) and RoBERTa (Jones et al., 2022), has revolutionized this field. These modern approaches, leveraging complex patterns in text, have shown superior performance over conventional stylometric techniques in identifying authorship (Saedi & Dras, 2021; Konstantinou et al., 2022; Valdez-Valenzuela et al., 2023). This paradigm shift underscores a significant evolution in AV methodologies, emphasizing the increasing effectiveness of machine learning in text analysis (Zheng & Jin, 2023).

While current AV models have made notable advancements, they predominantly focus on binary classification and notably lack in providing explanatory insights. Explainability is not only of academic interest. It is fundamental to understanding a model's decision-making logic, and it also enhances trust and reliability in the model's output. Additionally, the lack of clear explanations makes it hard to find and fix any biases that may be hidden inside these models, creating a significant problem for ensuring they are fair and unbiased. Therefore, it's critical for AI models not only to be accurate but also to provide transparency and interpretability.

This paper presents the InstructAV framework, an innovative approach tailored for AV tasks. Unlike existing models, InstructAV is designed to accurately verify authorship across texts while concurrently furnishing detailed linguistic explanations for its determinations.

---

[*]Equal contribution

A key feature of InstructAV is its unique capacity to integrate explainability directly into the classification process, thereby creating a direct pathway between making accurate predictions and offering deep explanations. Through rigorous testing across three diverse AV datasets, the InstructAV framework has demonstrated not only outstanding accuracy in authorship verification but has also set a new benchmark by producing coherent and substantiated explanations for its findings. This dual capability of InstructAV—merging high predictive performance with actionable insights—marks a significant leap forward in the AV domain, contributing both to the enhancement of model transparency and the advancement of explainable artificial intelligence.

Our contributions can be summarized as follows: (i) We propose the InstructAV framework for AV tasks to accurately determine whether two texts share the same author and to furnish robust linguistic explanations for the AV outcomes. (ii) We have curated three instruction-tuning datasets, each accompanied by dependable linguistic explanations for AV tasks. These datasets are intended to serve as valuable resources for advancing research in this field[1]. (iii) Both automated and human evaluation results demonstrate the effectiveness of the InstructAV in providing precise AV predictions and reliable linguistic explanations.

## 2 Related Work

### 2.1 Authorship Verification

In the last two decades, AV has evolved significantly, transitioning from traditional methods focusing on linguistic features like spelling and style to machine learning techniques (Boenninghoff et al., 2019). However, traditional machine learning, such as support vector machines, showed limited effectiveness (Konstantinou et al., 2022). Recent advancements involve contextual embeddings from language models like BERT, T5, and MPNET (Devlin et al., 2018; Raffel et al., 2020; Song et al., 2020), and further studies have explored graph convolutional networks and BiLSTM with attention mechanisms (Valdez-Valenzuela et al., 2023; Sun et al., 2023). Moreover, Huang et al. (2024) explored different representations of authorship to verify their effectiveness in encoding writing styles. They concluded that authorship representations might be expected to remain robust against certain types of data shifts.

Previous neural network-based methods, such as BERT and MPNET, have been crucial in advancing classification tasks, including AV. However, these models inherently lack mechanisms to elucidate their decision-making processes, a critical gap as the demand for explainability in AI grows Hung et al. (2023). Addressing this, Hung et al. (2023) introduced PromptAV, an innovative technique that leverages the capabilities of large language models (LLMs) for AV, employing step-by-step stylometric explanation prompts to enhance interpretability. Their findings demonstrate that PromptAV significantly outperforms traditional approaches like chain of thought (CoT) prompting (Wei et al., 2022) and PS+ prompting (Wang et al., 2023) in both accuracy and interpretability, marking a noteworthy advancement in the application of LLMs to AV tasks. Huang et al. (2024) utilized LLMs with the Linguistically Informed Prompting (LIP) technique for authorship verification, revealing that even without domain-specific fine-tuning, the LIP method guides LLMs to satisfactory performance.

Despite the progress, these methods' reliance on a few-shot demonstration model poses challenges in ensuring the consistency and relevance of their explanations. This limitation underscores a critical need for a more robust solution that can deliver both precise classification and meaningful explanations across a broader range of scenarios. To address this need, we introduce the InstructAV framework, which adopts an instruction fine-tuning approach to significantly enhance the classification accuracy and explanation quality in AV tasks. By refining the model's ability to generate relevant and consistent explanations, InstructAV not only builds on the foundation established by PromptAV but also addresses its primary limitations, offering a comprehensive solution that advances the field of explainable AV.

---

[1]The code and datasets can be found at `https://github.com/Social-AI-Studio/InstructAV`.

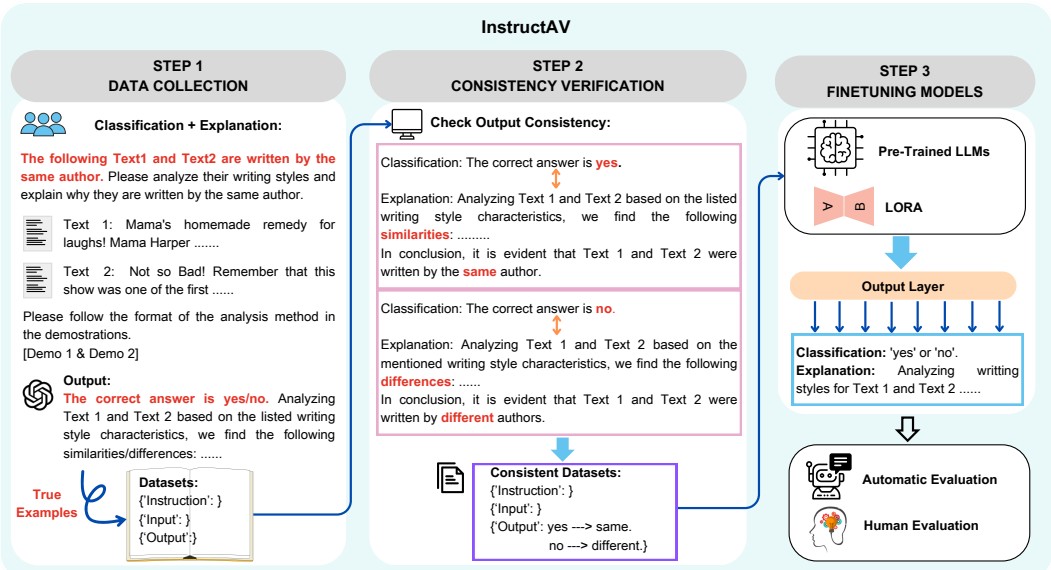

Figure 1: An illustration of InstructAV's architectures.

## 2.2 Parameter-Efficient Fine-Tuning of Large Language Models

The emergence of LLMs like GPT-3 represents a significant advancement in AI, expanding the capabilities of machine learning technologies. However, deploying these models poses challenges due to their high computational and memory demands. Parameter-Efficient Fine-Tuning (PEFT) addresses these issues. It selectively adjusts a small subset of model parameters, customizing it for specific downstream tasks with minimal resource usage Ding et al. (2023); Liu et al. (2022).

Within the PEFT paradigm, adapters are notable. These compact modules integrate into pre-trained networks, providing a resource-efficient way to customize models for specific tasks or datasets (Hu et al., 2023). By enabling targeted specialization without extensive retraining, adapters expand the accessibility of LLMs for various applications, enhancing their utility and flexibility.

The Low-Rank Adaptation (LoRA) adapter, a notable PEFT method, fine-tunes model weight matrices with low-rank modifications to boost task-specific trainability while retaining model strengths (Hu et al., 2021). Another method, Prefix-Tuning, adds task-specific parameters to transformer layers for alignment (Li & Liang, 2021). These methods merge adaptability with computational efficiency, enabling advanced LLMs to operate in resource-limited settings. LoRA, especially, preserves LLMs' general capabilities while enabling task-specific fine-tuning, marking a significant advancement in the field (Ding et al., 2023). In our study, we outperform AV tasks with LoRA fine-tuning.

## 3 Methodology

The overview of the InstructAV framework, as depicted in Figure 1, comprises three primary steps: data collection, consistency verification, and the fine-tuning of LLMs using the LoRA method (Hu et al., 2021). Initially, the framework focuses on the aggregation of explanatory data for AV samples. This approach uses the binary classification labels available in existing AV datasets. Following this, a strict quality check is implemented, aimed at verifying the alignment and consistency of the explanations with the corresponding classification labels. The final stage involves the synthesis of instruction-tuning data, which is a fusion of the gathered classification labels and their associated explanations. This composite data then serves as the foundation for fine-tuning LLMs in conjunction with the LoRA adaptation

technique. This approach ensures that the LLMs are not only accurately fine-tuned for the AV task but also enhanced in their capacity to provide coherent and reliable explanations for their predictions. The subsequent sections present the details of each component in InstructAV.

## 3.1 Explanation Data Collection

To augment the explanatory capabilities of the InstructAV model, particularly in generating dependable explanations for AV predictions, we initiated a comprehensive collection of explanations using ChatGPT, with the crucial step of informing ChatGPT about the classification labels beforehand. This process involved three datasets widely used in AV studies: the IMDB dataset (Seroussi et al., 2014), the Twitter dataset (Schwartz et al., 2013), and the Yelp Reviews dataset[2]. This selection strategically covers various dimensions of textual data, thereby ensuring a diverse and comprehensive analysis.

The IMDB dataset, characterized by its longer text length (averaging 303 words), exemplifies long-form content. In contrast, the Twitter dataset, with an average text length of 16 words, epitomizes short-form content. The Yelp Reviews dataset, averaging 154 words per text, represents medium-length content. These datasets, initially curated for authorship attribution tasks, were adapted for our study. We extracted 11,000 samples from each dataset to facilitate a robust AV evaluation. Each sample comprises two texts, either written by the same author or by different authors.

For generating explanations, we employed ChatGPT with true classification labels and few-shot prompts, focusing on 11 linguistic features for each AV sample. These features, identified in the research by (Boenninghoff et al., 2019), are crucial for analyzing writing styles in textual content. The linguistic features generated by known-label ChatGPT were recorded as labels for the AV explanations in our dataset if they accurately predict and meet the consistency check.

To link classification labels closely with linguistic explanations during the data collection phase, we crafted prompts that incorporate the classification labels. For example, a prompt might state "*The following Text1 and Text2 are written by different authors. Please analyze their writing styles and explain why they are written by different authors.*" to guide the explanation process. A detailed example of the explanation generation prompt is shown in Appendix B.2 Table 8. These prompts enhance the relevance of the collected explanations, thereby improving the explanatory capacity of the InstructAV model.

**Consistency Verification.** Alignment between classification labels and their linguistic explanations is essential for explanation data integrity and trustworthiness. Models like known-label ChatGPT are skilled at generating explanations but face challenges in align explanations with classification labels. For instance, despite being informed that "*The following Text1 and Text2 are written by the same author,*" ChatGPT might incorrectly respond with, "*The correct answer is no.*" Instances like this create a mismatch between explanations and classification labels, reducing trust in the model's explanations. To enhance user trust in automated decisions, it is important to guarantee the consistency and reliability of both classifications and explanations. As shown in Figure 1, we employ a consistency verification method to verify the alignment between the model's analytical explanations and its classification decisions. We have instituted a comprehensive verification process to ensure the explanations are consistent with the classification. This process leverages demonstration templates that inherently guide the model to incorporate specific expressions—such as '*same/similarities*' or '*different/differences*'—into its outputs. During the consistency verification stage, keyword searching was performed on terms within the generated text. Matching the phrases '*written by the same author*' (resp. '*written by different authors*') with classification labels allows us to assess the quality of ChatGPT's linguistic explanations.

The key to this process is the construction of instruction-tuning data, which serves a dual purpose: facilitating AV classification prediction and supporting the generation of explanations. An example of the instruction-tuning data is shown in Appendix B.3 Table 9. This

---

[2]https://www.yelp.com/dataset

| Dataset | #Authors | #Train | #Test | Avg length |
|---------|----------|--------|-------|------------|
| IMDB | 64 | 10,000 | 1,000 | 303 |
| Twitter | 100 | 10,000 | 1,000 | 16 |
| Yelp | 1,000 | 10,000 | 1,000 | 154 |

Table 1: Dataset statistics for IMDB, Twitter and Yelp.

| Dataset Setting | Model | IMDB | Twitter | Yelp |
|-----------------|-------|------|---------|------|
| | BERT | 0.677 (0.0124) | 0.702 (0.0021) | 0.622 (0.0020) |
| | DistilBERT | 0.526 (0.0125) | 0.575 (0.0065) | 0.543 (0.0036) |
| | ALBERT | 0.642 (0.0040) | 0.701 (0.0023) | 0.601 (0.0020) |
| | LIP (GPT-4-turbo) | 0.732 (0.0070) | 0.612 (0.0040) | 0.632 (0.0036) |
| Classification | LIP (LLaMA-2-70B) | 0.533 (0.0080) | 0.554 (0.0042) | 0.528 (0.0040) |
| | LIP (Mistral-7B) | 0.507 (0.0038) | 0.539 (0.0025) | 0.527 (0.0032) |
| | InstructAV (LLaMA-1-7B) | 0.648 (0.0236) | 0.610 (0.0062) | 0.542 (0.0031) |
| | InstructAV (OPT-6.7B) | 0.590 (0.0050) | 0.524 (0.0110) | 0.527 (0.0060) |
| | InstructAV (LLaMA-2-7B) | **0.914** (0.0046) | **0.740** (0.0070) | **0.689** (0.0025) |
| | PromptAV-2shot (GPT-3.5) | 0.623 (0.0397) | 0.628 (0.0147) | 0.534 (0.0064) |
| | PromptAV-4shot (GPT-3.5) | 0.635 (0.0265) | 0.667 (0.0163) | 0.544 (0.0080) |
| Classification & | PromptAV-8shot (GPT-3.5) | 0.601 (0.0070) | 0.648 (0.0075) | 0.564 (0.0081) |
| | PromptAV (GPT-4-Turbo) | 0.755 (0.0075) | 0.729 (0.0070) | 0.597 (0.0065) |
| Explanation | InstructAV (LLaMA-1-7B) | 0.825 (0.0289) | 0.625 (0.0065) | 0.596 (0.0104) |
| | InstructAV (OPT-6.7B) | 0.744 (0.0095) | 0.714 (0.0070) | 0.575 (0.0140) |
| | InstructAV (LLaMA-2-7B) | **0.937** (0.0017) | **0.745** (0.0063) | **0.693** (0.0442) |

Table 2: Classification Accuracy of InstructAV and the baselines on different dataset settings. Highest Acc are **bolded**. All experiments were repeated three times. The table lists the average values of the three repetitions, while the standard deviations of accuracy from the experiments are provided in brackets.

verification step is essential for ensuring that the InstructAV model is not only accurate in its predictions but also capable of generating explanations that are coherent, relevant, and aligned with the classification outcomes, thereby enhancing the overall efficacy and reliability of the InstructAV framework.

## 3.2 Fine-tuning with LoRA

Adapting LLMs for AV tasks can be a demanding process, particularly due to the significant computational resources and extensive labeled data typically required for fine-tuning. To address this challenge, we have incorporated a PEFT method known as LoRA (Hu et al., 2021) into our approach for adapting LLMs to AV tasks.

LoRA presents a novel approach to updating weight matrix parameters. It decomposes the matrix into two low-rank matrices, reducing trainable parameters while processing high-dimensional matrices effectively. During the forward pass, LoRA matrices compute rank-decomposed weights, used in attention or feed-forward activations. Importantly, this method keeps the LLM's core parameters unchanged, enabling adaptation to new data distributions or tasks without overhauling the model completely.

The LoRA architecture is encapsulated in the following equation:

$$h = W_0 x + \Delta W_0 x = W_o x + BAx \qquad (1)$$

where $W_0 \in \mathbb{R}^{d \times d}$ is a pre-trained weight matrix in LLMs, which is associated with two adjustable low-rank matrices, $B \in \mathbb{R}^{d \times r}$ and $A \in \mathbb{R}^{r \times d}$, $d$ corresponds to the hidden dimension of the attention layers, and $r$ is the adaptation rank, which is selected such that $r \ll d$. Within the training phase, $W_0$ remains frozen, exempt from gradient updates, whereas $A$ and $B$ are dynamic, containing the parameters subject to training.

# 4 Experiments

## 4.1 Experiment Settings

**Datasets.** The InstructAV framework evaluation utilized three distinct Authorship Verification (AV) datasets: IMDB62 (Seroussi et al., 2014), Twitter (Schwartz et al., 2013), and Yelp Reviews [3], chosen for diversity and relevance.

To evaluate the AV classification and explanation tasks performance, we constructed two distinct types of dataset settings, each incorporating varying levels of information:

1. *Classification*: This dataset involves integrating a question alongside two texts as the input and employs the LoRA method to fine-tune the model for classification tasks. The expected output is a straightforward binary classification indicating whether the two texts are written by the same author, formatted as "*The correct answer is yes/no*".

2. *Classification and Explanation*: In this setting, we augment the classification data with linguistic analysis to empower the model to generate robust explanations for the AV predictions. The LLMs are fine-tuned to not only predict the classification labels, but also provide an analysis of eleven linguistic features of the two texts. This added layer of analysis offers a reasoned explanation behind the classification decision, thereby enhancing the model's interpretability and reliability.

For the *Classification* setting, We randomly sampled 11,000 balanced samples from the IMDB62, Twitter, and Yelp datasets, respectively. The data was then divided into a training set comprising 10,000 samples and a test set comprising 1,000 samples, with both sets maintaining balanced class distributions.

For the *Classification and Explanation* setting, we initially sampled 20,000 examples from each dataset and used the GPT-3.5-turbo API for linguistic analysis. To ensure the high quality of the generated explanations, samples with incorrect linguistic analysis were dropped during the initial consistency verification phase. For a comprehensive evaluation, balanced subsets consisting of 10,000 training samples and 1,000 testing samples were randomly selected from the verified examples within each subset. These samples formed instruction-tuning data, including text and corresponding linguistic explanations (see Section 3.1). Details regarding dataset statistics and characteristics, including the splits between training and test sets, are presented in Table 1.

These two dataset settings enable a thorough investigation into how the addition of explanatory components influences the performance of the classification task, providing insights into the efficacy and adaptability of the InstructAV framework.

**Baselines.** For the *Classification* task, our baseline models comprised BERT (Devlin et al., 2018) and its variants, DistilBERT (Sanh et al., 2019), and AlBERT (Lan et al., 2019). These models were selected for their widespread usage and proven effectiveness in AV classification tasks (Brad et al., 2021; Tyo et al., 2022; Fabien et al., 2020). However, these models are not inherently designed for text generation tasks. Another baseline related to LLMs is the application of the LIP method proposed by Huang et al. (2024) on GPT-4-turbo, LLaMA-2-70B, and Mistral-7B.

For tasks involving *Classification and Explanation*, autoregressive models like GPT, which generate text sequences based on preceding text, are suitable. We adopted the PromptAV methodology on GPT-3.5-Turbo as a baseline for the ChatGPT model, as proposed by (Hung et al., 2023). Although PromptAV may be slightly less effective than BERT in classification accuracy, it offers advantages such as bypassing extensive training and generating explanations. Additionally, we integrated PromptAV, using GPT-4 as the foundational model, to establish a more competitive baseline.

---

[3]https://www.yelp.com/dataset

| IMDB | | | | |
|---|---|---|---|---|
| **Model** | ROUGE-1 | ROUGE-2 | ROUGE-L | Bert_Score |
| PromptAV-2shot (GPT-3.5) | 0.379 | 0.147 | 0.227 | 0.844 |
| PromptAV (GPT-4) | 0.496 | 0.193 | 0.237 | 0.861 |
| InstructAV (LLaMA-1) | 0.677 | 0.412 | 0.496 | 0.898 |
| InstructAV (OPT) | 0.656 | 0.403 | 0.482 | 0.893 |
| InstructAV (LLaMA-2) | **0.689** | **0.434** | **0.515** | **0.907** |
| Twitter | | | | |
| PromptAV-2shot (GPT-3.5) | 0.445 | 0.203 | 0.287 | 0.856 |
| PromptAV (GPT-4) | 0.510 | 0.307 | 0.466 | 0.860 |
| InstructAV (LLaMA-1) | 0.670 | 0.406 | 0.522 | 0.899 |
| InstructAV (OPT) | 0.644 | 0.398 | 0.476 | 0.897 |
| InstructAV (LLaMA-2) | **0.689** | **0.420** | **0.542** | **0.904** |
| Yelp | | | | |
| PromptAV-2shot (GPT-3.5) | 0.377 | 0.142 | 0.243 | 0.856 |
| PromptAV (GPT-4) | 0.431 | 0.162 | 0.402 | 0.867 |
| InstructAV (LLaMA-1) | 0.666 | 0.407 | 0.524 | 0.906 |
| InstructAV (OPT) | 0.629 | 0.377 | 0.486 | 0.900 |
| InstructAV (LLaMA-2) | **0.716** | **0.429** | **0.592** | **0.912** |

Table 3: Automatic Metric Performance for InstructAV and PromptAV. Higher Acc are **bolded**.

Our comparative analysis evaluates both classification accuracy and the quality of explanations from InstructAV and PromptAV. This comprehensive assessment aims to understand the strengths and potential limitations of each approach in AV tasks.

**Implementation.** In our research, we employed three different LLMs to validate the efficacy of the InstructAV framework. These models include OPT-6.7B (Zhang et al., 2022), LLaMA-1-7B (Touvron et al., 2023a), and LLaMA-2-7B (Touvron et al., 2023b). Our experiments were conducted using two NVIDIA A6000 GPUs, encompassing both the fine-tuning and inference phases.

For fine-tuning, we applied the LoRA approach with a rank set to 8. This introduced 4.1 million parameters, representing only 0.06% of the total 7 billion parameters of the base models. This parameter increase highlights LoRA's effectiveness in offering an efficient fine-tuning method. Fine-tuning was performed over 3 epochs using the LLM-Adapters Toolkit (Hu et al., 2023), tailored for integrating adapters into LLMs.

During inference, all operations were performed deterministically with a fixed temperature of 0.1. This maintains consistency and reliability in the model's performance, crucial for accurately assessing the InstructAV framework's capabilities. The selection of diverse LLMs and the strategic application of LoRA for fine-tuning effectively showcase the capability of the InstructAV framework across various model architectures. To ensure the reproducibility of the experiments conducted using the InstructAV framework, each experiment was repeated three times with different random seeds. The mean and standard deviation were calculated from these repetitions.

## 4.2 Evaluation Metrics

The evaluation of the InstructAV framework involves both AV classification and explanation generation tasks. This dual-focused approach enables an in-depth assessment of the framework's capabilities in key areas of AV.

For AV classification tasks, accuracy serves as the primary metric. This metric effectively measures the models' proficiency in determining whether two texts have been written by the same author. To evaluate the framework's performance in generating explanations for

| IMDB | | | | |
|---|---|---|---|---|
| Model | Coverage | Relevance | Reasonableness | Persuasiveness |
| Explanation Label | 11 | 4.42 | 4.12 | 4.22 |
| PromptAV-2shot (GPT-3.5) | 7 | 3.8 | 3.84 | 3.26 |
| PromptAV (GPT-4) | 7 | 3.92 | 3.87 | 3.56 |
| InstructAV (LLaMA-2-7B) | 11 | 4.24 | 3.98 | 4.03 |
| Twitter | | | | |
| Explanation Label | 11 | 4.66 | 4.42 | 4.65 |
| PromptAV-2shot (GPT-3.5) | 7 | 3.01 | 3.24 | 2.94 |
| PromptAV (GPT-4) | 7 | 3.67 | 3.22 | 3.27 |
| InstructAV (LLaMA-2-7B) | 11 | 4.48 | 4.25 | 4.28 |
| Yelp | | | | |
| Explanation Label | 11 | 4.56 | 4.44 | 4.65 |
| PromptAV-2shot (GPT-3.5) | 7 | 2.87 | 3.02 | 3.06 |
| PromptAV (GPT-4) | 7 | 3.75 | 3.41 | 3.50 |
| InstructAV (LLaMA-2-7B) | 11 | 4.3 | 4.32 | 4.36 |

Table 4: Results of the human evaluation on explanation labels and explanations generated by the InstructAV and PromptAV frameworks.

AV tasks, a distinct approach is required. Here, the focus shifts towards assessing the quality of the linguistic analysis generated by the models. Unlike accuracy, which is quantitatively measurable, the quality of linguistic explanations presents a more complex evaluation challenge, as it involves subjective elements related to the coherence, relevance, and clarity of the explanations. Given the complex nature of evaluating explanation quality, we adopt a dual evaluation approach, employing both automatic and human evaluation metrics.

**Automatic Evaluation for Explanations.** In our automated assessment of explanation quality for both InstructAV models and baseline models, we measure the similarity between generated and labeled explanations, using ChatGPT-generated texts as standards. Through consistency verification, we ensure explanatory labels within test sets consistently align with classification labels, affirming their accuracy. Consequently, we consider ChatGPT's linguistic analysis as explanation labels. We employ metrics like ROUGE-1, ROUGE-2, ROUGE-L (for content coverage and structural fluency), and BERT_Score (for semantic quality). Higher scores indicate better explanation quality, ensuring a comprehensive evaluation based on content accuracy, logical coherence, and contextual relevance.

**Human Evaluation for Explanations.** To supplement the automated metrics for evaluating explanation quality in the InstructAV framework, we also implemented a human evaluation. This approach involved randomly selecting 100 examples from each of the IMDB, Twitter, and Yelp Reviews datasets for qualitative assessment by human evaluators. These evaluations included not only explanations generated by InstructAV and baseline models, but also those explanation labels. For the human evaluations focusing on explanations, we selected InstrctAV (LLaMA-2-7B) along with the two most robust baselines, namely PromptAV-2shot equipped with GPT-3.5 and GPT-4 as the base models. Three evaluators were enlisted to independently evaluate each explanation across four criteria:

**Coverage**: Evaluating how many of the anticipated linguistic features are present in the explanations. Evaluators checked the number of features covered in each explanation, with baseline models like PromptAV covering 0 to 7 features, and InstructAV, along with explanation labels, extending from 0 to 11 features.

**Relevance**: Evaluators scored the relevance of the explanations to the original texts on a 5-point Likert scale, ranging from 1 (completely irrelevant) to 5 (perfectly relevant).

**Reasonableness**: This involved assessing the logical soundness of each feature's explanation, using a 5-point Likert scale, from 1 (totally unreasonable) to 5 (completely reasonable).

**Persuasiveness**: Evaluators rated the persuasiveness of the explanations on a scale from 1 (completely not persuasive) to 5 (highly persuasive).

4.3   Experiment Results

**Classification Results.** We evaluate the InstructAV framework for AV tasks, utilizing both classification and explanatory dataset settings. The primary goal is to explore how linguistic analysis can improve the model's performance in AV classification tasks.  The results, detailed in Table 2, include the accuracy of InstructAV with various LLMs and baseline models. A key finding is that InstructAV, particularly when paired with LLaMA-1-7B and LLaMA-2-7B, outperforms baseline models in all datasets when using only classification data. Notably, with LLaMA-2-7B, InstructAV achieves a 25.2% improvement over the highest-performing baseline, BERT, on the IMDB dataset.

It's imperative to underline that in our experimental setup, models were evaluated on their ability to concurrently generate classification predictions and conduct linguistic analysis without incorporating explanation data in the input. This methodology ensured fair comparison with approaches that focus only on classification. Our results clearly demonstrate that all variants of the InstructAV framework, utilizing different LLMs, significantly benefit from training on explanatory labels.  Notably, InstructAV with LLaMA-2-7B showcased a remarkable 27.1% improvement in classification accuracy over the PromptAV approach using 2-shot prompts based on GPT-3.5. Furthermore, when compared to PromptAV-2shot employing GPT-4, InstructAV exhibited superior performance across all three evaluated datasets.  These findings highlight the substantial benefits of incorporating explanatory training, indicating that providing InstructAV with the dual function of AV classification and generating linguistic explanations significantly boosts its classification precision.

**Automatic Evaluation Results on Explanations.** Our InstructAV framework demonstrates that training in explanation labels can significantly improve AV classification performance. Additionally, we aim to assess whether InstructAV can generate high-quality explanations. For this purpose, we selected PromptAV-2shot (GPT-3.5), the best-performing variation of the framework, alongside GPT-4, to benchmark against all variations of InstructAV. we subjected them to both automatic and human evaluations focused on the quality of their generated linguistic explanations. The results of the automatic evaluation are presented in Table 3. These results reveal that InstructAV consistently surpasses PromptAV models (GPT-3.5 and GPT-4) across all datasets and all evaluation metrics. Notably, ROUGE-1 and ROUGE-2 scores highlight InstructAV's superior performance in achieving content overlap at both the word and phrase levels. Moreover, the ROUGE-L metric indicates that InstructAV is more proficient in maintaining sentence-level structure and fluency. Furthermore, the BERT_Score supports the observation that the explanations generated by InstructAV are semantically closer to the explanation labels. This comprehensive evaluation underscores InstructAV's capability not only in improving AV classification accuracy but also in generating linguistically coherent and contextually relevant explanations.

**Human Evaluation Results on Explanations.** To comprehensively evaluate the generated explanations and to assess the quality of the linguistic analysis produced by ChatGPT, which serves as our explanation labels, we have conducted a human evaluation using four key metrics: Coverage, Relevance, Reasonableness, and Persuasiveness. The results of this human evaluation are presented in Table 4. Our findings from this human evaluation process show that our explanation labels achieves the highest scores. This result validates our methodological choice of using explanations generated by known-label ChatGPT as both a source for training data and a benchmark for explanation labels in our testing scenarios. Importantly, the results also reveal that InstructAV, particularly with the LLaMA-2-7B model, not only surpasses the performance of PromptAV-2shot models (GPT-3.5 and GPT-4) but also attains a level of explanation quality that is comparable to known-label ChatGPT. This outcome is significant as it demonstrates that InstructAV can produce explanations that are not only accurate but also contextually relevant, logically sound, and convincing to human evaluators. Such a capability is essential for applications where understanding the rationale behind model predictions is as important as the predictions themselves.

**Correlation between Explanation and Classification.** To explore the relationship between explanation quality and classification accuracy, we have selected two distinct subsets of InstructAV samples: the top 25% with the highest average human evaluation scores and the bottom 25% with the lowest average human evaluation scores. We then calculated the

|  | IMDB | Twitter | Yelp |
|---|---|---|---|
| Top 25 | **0.92** | **0.8** | **0.84** |
| Bottom 25 | 0.88 | 0.68 | 0.72 |

Table 5: Performance of InstructAV on Top 25 and Bottom 25 explanation. Higher accuracies are bolded.

|  | Model | IMDB | Twitter | Yelp |
|---|---|---|---|---|
| | LLaMA-1-7B | 0.000 | 0.000 | 0.000 |
| 0-shot | OPT-6.7B | 0.000 | 0.000 | 0.000 |
| | LLaMA-2-7B | 0.006 | 0.010 | 0.021 |
| | LLaMA-1-7B | 0.189 | 0.097 | 0.226 |
| 2-shot | OPT-6.7B | 0.258 | 0.179 | 0.297 |
| | LLaMA-2-7B | 0.309 | 0.336 | 0.397 |
| | LLaMA-1-7B | 0.014 | 0.020 | 0.127 |
| 4-shot | OPT-6.7B | 0.191 | 0.220 | 0.217 |
| | LLaMA-2-7B | 0.279 | 0.350 | 0.375 |
| | LLaMA-1-7B | 0.003 | 0.007 | 0.002 |
| 8-shot | OPT-6.7B | 0.005 | 0.008 | 0.003 |
| | LLaMA-2-7B) | 0.020 | 0.033 | 0.025 |
| | InstructAV (LLaMA-1-7B) | 0.648 | 0.610 | 0.542) |
| Finetuned | InstructAV (OPT-6.7B) | 0.590 | 0.524 | 0.527 |
| | InstructAV (LLaMA-2-7B) | **0.914** | **0.740** | **0.689** |

Table 6: Classification Performance of InstructAV Framework and LLMs in Few-Shot Contexts Without Fine-Tuning. Highest Acc are **bolded**.

classification accuracy for each of these subsets, and the results are presented in Table 5. Our analysis clearly indicates that samples associated with higher quality explanations consistently achieve superior classification accuracy. This finding underscores the effectiveness of training InstructAV to not only generate AV classification predictions but also provide linguistic explanations. Such training not only enhances the model's performance in AV classification tasks but also enhances its ability to produce valuable linguistic analysis.

## 5 Ablation Study

Ablation experiments were conducted to investigate potential data contamination issues, specifically whether the data might have been included in the training corpora of LLMs. We performed 0-shot, 2-shot, 4-shot, and 8-shot in-context tests on each dataset using the original, untuned LLaMA-2-7B, LLaMA-1-7B, and OPT-6.7B models. Each experiment was replicated three times, with the mean results presented in Table 6. The results indicate that the original models struggle to perform AV tests effectively on our dataset, especially in the 0-shot setting where the outputs are highly random. Both LLaMA-1-7B and OPT-6.7B failed to make correct judgments, and LLaMA-2-7B showed weak judgment capabilities. Consequently, we postulate that it is unlikely that our dataset was included in the training data of the original models.

## 6 Conclusion

This research presents InstructAV, an innovative approach to AV tasks that leverages LLMs with a PEFT method. Our study establishes InstructAV as a significant advancement in the AV domain, showcasing its ability to enhance classification accuracy and provide clear and coherent explanations for its decisions. The contributions of this paper, including the development of the InstructAV framework, the creation of three instruction-tuning datasets with reliable linguistic explanations, and the demonstration of the framework's effectiveness through both automated and human evaluations, mark a crucial progress in AV research. InstructAV, with its dual priority on high accuracy and the ability to provide high quality explanations, positions it as a state-of-the-art AV solution.

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

# A    Limitations

The InstructAV framework, while achieving state-of-the-art performance in AV classification and explanation tasks, does face a notable limitation in its current form. The human evaluations of model-generated explanations differ from typical evaluations in other tasks. Specifically, our evaluations are intended for high-level verification to ensure that the explanations produced by ChatGPT are coherent and align with the text , not as a full assessment of authorial features. Author analysis involves critical frequency and nuanced features better captured by computational methods then humans (Stamatatos et al., 1999). Theoretically, an ideal human evaluation would ask human participants to justify why two texts are written by the same author, but such research is costly and problematic, as humans do not perform well in tasks that require analysis of writing styles. Considering these limitations, we consider ChatGPT-generated explanations to be a reasonable and cost-effective alternative. Future work will consider a more suitable evaluation method that incorporates computational features.

On the other hand, when tasked with generating both AV classification predictions and linguistic explanations, InstructAV is constrained to produce a maximum of 512 new tokens. This restriction can lead to significantly longer inference times. For example, generating explanations for 1,000 samples in a long-text dataset like IMDB can take around 4-5 days, while in shorter-text datasets such as Twitter, this time is reduced to approximately 2 days. Addressing this challenge, future work will focus on developing methods to enhance the efficiency of the inference process within InstructAV, particularly in the context of generating explanations.

# B    Appendix

## B.1    Case Study for InstructAV

We have included randomly selected examples to showcase the capabilities of InstructAV in generating both classification predictions and language feature-based explanations. One such example from the IMDB dataset is detailed in Table 7. This example demonstrates how InstructAV first provides a classification prediction and then follows it with detailed explanations anchored in specific language features pertinent to the classification.

These examples highlight InstructAV's proficiency in delivering accurate classifications paired with clear and credible explanations. Notably, the explanations are tailored to each specific sample, rather than being generic or broad. This customization enhances the explanations' effectiveness, as they are directly correlated with the content of the original text. For instance, the model explicates features such as writing style and use of expressions by incorporating words and phrases directly from the text in question. This approach not only bolsters the relevance of the explanations but also their trustworthiness, providing users with insightful and contextually grounded rationale behind each classification decision made by InstructAV.

## B.2    Prompt Templates

## B.3    Collected Datasets

**Instruction**: *Please decide if the following Text1 and Text2 are written by the same authors. 'yes' means from the same author, 'no' means not from the same author. Then, provide an analysis based on writing styles.*

**Text 1:** *The British Soap Awards 2005 I am a fan of only two soaps , Coronation Street and EastEnders , but I am always going to watch this programme to see what the other soaps are like , if they deserve to win something , and what I may be missing out on . I always want to agree with the awards ...* [Truncated due to length restriction]

**Text 2:** *Father of the Bride As far as I can work out , it is after this film that the career of Steve Martin has had very few laughs , which is a shame , he just needs to find the right film . Anyway , this is the remake of the 1950 film , and it sees middle aged father George Banks ( Martin ) with wife Nina ( Diane Keaton ) having to prepare the wedding for their 21-year-old daughter Annie Banks ( Kimberly Williams-Paisley ) ...* [Truncated due to length restriction]

**Generated Output**

**Classification:** The correct answer is yes.

**Explanation:** Upon analyzing Text 1 and Text 2 based on the listed writing style characteristics, we find the following similarities that could suggest that both texts were written by the same author:

**Writing Style:** Both texts provide personal opinions and evaluations of the subject matter. Text 1 focuses on the British Soap Awards, while Text 2 provides a review of the film \"Father of the Bride.\""

**Expressions and Idioms:** Both texts use colloquial expressions and idioms, such as \"if they deserve to win something\" in Text 1 and \"he just needs to find the right film\" in Text 2.

**Tone and Mood:** Both texts have an informal and conversational tone. The author expresses their personal preferences and opinions in both texts.

**Sentence Structure and Syntax:** Both texts use a mix of compound and simple sentences. For example, \"I always want to agree with the awards that are given according to the clips/photos shown\" in Text 1 and \"Martin does do well, Keaton offers a good support, and Short adds giggles as the foreign wedding organiser\" in Text 2.

**Punctuation Style:** Both texts use consistent punctuation, including commas and full stops.

**Special Characters Style, Capitalization Style:** Both texts capitalize proper nouns and show titles. However, Text 2 uses unconventional capitalization for emphasis, such as \"Very good!\"

**Compound and Separate Spelling:** No significant differences in spelling patterns are observed between the two texts.

**Acronyms and Abbreviations:** Both texts use acronyms and abbreviations, such as \"EE\" for \"EastEnders\" in Text 1 and \"B.D.Wong\" for \"B.D. Wong\" in Text 2.

**Characters Style:** Both texts mention specific actors and their roles, indicating a familiarity with the subject matter.

**Diatopic Variations and Foreign Languages:** Both texts appear to be written in standard English without diatopic variations or foreign language references.

**Other Relevant Aspects:** Both texts provide specific details about the subject matter, indicating a personal familiarity or interest in the topic. The author's opinions and evaluations are evident in both texts.

Table 7: A text pair example with the generated responses from InstructAV

**Task**: Text1 and Text2 are written by the same author. Please analyze their writing styles and explain why they are written by the same author. You can refer to the following characteristics of writing style. 1. writing style. 2. expressions and Idioms. 3. tone and mood. 4. sentence structure and syntax. 5. punctuation style. 6. special characters style, capitalization style. 7. compound and separate spelling. 8. acronyms and abbreviations. 9. characters style. 10. Diatopic variations and foreign languages. 11. any other relevant aspect.

**Text 1:** *The British Soap Awards 2005 I am a fan of only two soaps , Coronation Street and EastEnders , but I am always going to watch this programme to see what the other soaps are like , if they deserve to win something , and what I may be missing out on. I always want to agree with the awards that are given according to the clips / photos shown , and that is what I also enjoy , watching back on the past year ...* [Truncated due to length restriction]

**Text 2:** *Father of the Bride As far as I can work out , it is after this film that the career of Steve Martin has had very few laughs , which is a shame , he just needs to find the right film. Anyway , this is the remake of the 1950 film , and it sees middle aged father George Banks ( Martin ) with wife Nina ( Diane Keaton ) having to prepare the wedding for their 21-year-old daughter Annie Banks ( Kimberly Williams-Paisley )....* [Truncated due to length restriction]

Please follow the format of the analysis method in the demostrations. You will be given 2 demostrations.

### **Demostration Start:**

Text 1: Mama's homemade remedy for laughs! Mama Harper develops her mother's concoction to help her family get over an illness like the flu. After her son and bimbo-in-law Naomi who is now expecting their child realized that they can't afford it. They try a get rich scheme which falls on the unmentioned ingredient , alcohol......[Truncated due to length restriction]

Text 2:Not so Bad! Remember when this show was one of the first sitcoms in syndication in the 1980s. It was a new thing. When a sitcom or drama could not get a network , they went with syndication. I thought this show was alright in syndication because it reached a wide audience. I often saw it on Saturday evenings usually before Mama's Family. I was a kid so I didn't know better......[Truncated due to length restriction]

Upon analyzing Text 1 and Text 2 based on the listed writing style characteristics, we find the following similarities that could suggest that both texts were written by the same author:

Writing Style: Both texts seem to be reminiscing about older television shows. The author uses a mixture of narrative and opinion in both texts, indicating a personal connection or memory attached to the subject matter.

Expressions and Idioms: The expressions "I miss this show" and "I don't recall ever not watching it" showcase a personal touch and sentimentality in both texts.

Tone and Mood: Both texts have a nostalgic and somewhat informal tone. The author shares memories and impressions about shows from the past, using phrases like "I was a kid so I didn't know better" and "Still, I miss this show."

Sentence Structure and Syntax: Both texts employ a mix of compound and simple sentences. For instance, "When a sitcom or drama could not get a network, they went with syndication" and "Mama Harper develops her mother's concoction to help her family get over an illness like the flu."

Punctuation Style: Both texts employ spaces before and after commas inconsistently, such as in "$180 , 000" and "Marla Pennington , Edie McClurg."

Special Characters Style, Capitalization Style: Both texts capitalize show titles like "Mama's Family" and show proper nouns. However, both texts sometimes lack proper capitalization at the start of sentences, such as "Still" in Text 1.

Compound and Separate Spelling: No significant differences in spelling patterns are observed between the two texts. Acronyms and Abbreviations: The second text uses "1980s" to denote the decade, showcasing a preference for this kind of abbreviation.

Characters Style: The character portrayal in both texts has a casual undertone. For example, "bimbo-in-law Naomi" and "the guy who played her father."

Diatopic Variations and Foreign Languages: Both texts appear to be written in American English and reference American television shows.

Other Relevant Aspects:Both texts contain anecdotal or subjective experiences like "I was a kid so I didn't know better" and "Still, I miss this show." The author provides specific details about show characters, indicating a personal familiarity with the subject. There's a recurring theme of syndication and older television programming in both texts, further suggesting the same author might have written them.

In conclusion, based on the aforementioned similarities in writing style, expressions, tone, and other characteristics, it is plausible that Text 1 and Text 2 were written by the same author.

Table 8: Prompt used for generating explanation labels

**Classification Datasets**

**Instruction**: Please decide if the following Text 1 and Text 2 are written by the same authors. 'yes' means from the same author, 'no' means not from the same author.

**Text 1**: *Send in the Clowns!!! I don't know why we haven't heard of Judy Cohen since this episode. She can break your heart singing Stephen Sondheim's Send in the Clowns. She plays herself as an aspiring singer who beats herself down very much because she is overweight and not the classic beauty......*[Truncated due to length restriction]

**Text 2**: *Could have been better! Columbo's Most Dangerous Match is about chess. An unlikely subject or motive for murder in my opinion. The star is Manchurian Candidate's Lawrence Harvey who plays a deaf chess champion......*[Truncated due to length restriction]

**Output**: The correct answer is yes.

**Classification & Explanation Datasets**

**Instruction**: *Please decide if the following Text 1 and Text 2 are written by the same authors. 'yes' means from the same author, 'no' means not from the same author. Then, provide an analysis based on writing styles.*

**Text 1**: *Send in the Clowns!!! I don't know why we haven't heard of Judy Cohen since this episode. She can break your heart singing Stephen Sondheim's Send in the Clowns. She plays herself as an aspiring singer who beats herself down very much because she is overweight and not the classic beauty......*[Truncated due to length restriction]

**Text 2**: *Could have been better! Columbo's Most Dangerous Match is about chess. An unlikely subject or motive for murder in my opinion. The star is Manchurian Candidate's Lawrence Harvey who plays a deaf chess champion......*[Truncated due to length restriction]

**Output**: The correct answer is yes. Analyzing Text 1 and Text 2 based on the listed writing style characteristics, we find the following similarities that could suggest that both texts were written by the same author:

Writing Style: Both texts provide detailed commentary and analysis of specific episodes of a TV show. The author shares personal opinions and emotional responses to the episodes.

Expressions and Idioms: Both texts use phrases like "break your heart" and "a hell of a game" to evoke emotional reactions and convey personal opinions.

Tone and Mood: Both texts exhibit enthusiasm and engagement with the subject matter. The author expresses admiration for certain performances and emphasizes their impact.

Sentence Structure and Syntax: Both texts use a mix of simple and compound sentences. The sentences focus on describing specific events and interactions between characters.

Punctuation Style: Both texts use consistent punctuation, including proper use of commas and ellipses. They enhance the flow and expressiveness of the narrative.

Special Characters Style, Capitalization Style: Both texts consistently capitalize proper nouns and show titles. The author pays attention to the details and presents them accurately.

Compound and Separate Spelling: Both texts use standard spelling patterns without significant variations.

Acronyms and Abbreviations: Neither text employs acronyms or abbreviations.

Characters Style: Both texts describe specific characters and their roles within the TV show episodes. The author demonstrates familiarity with the characters' interactions and motivations.

Diatopic Variations and Foreign Languages: Both texts are written in English without reference to specific geographical locations or foreign languages.

Other Relevant Aspects: Both texts contain personal appeals to specific individuals, such as "Where are you Judy Cohen?" and "You almost feel sorry for him." The author shows a deep connection to the subject matter and a desire for continued engagement.

In conclusion, based on the aforementioned similarities in writing style, expressions, tone, and other characteristics, it is likely that Text 1 and Text 2 were written by the same author.

Table 9: Example of collected **Classification** Datasets and **Classification & Explanation** Datasets

