# OpenReview forum: "InstructAV: Instruction Fine-tuning Large Language Models for Authorship Verification"
_colmweb.org/COLM/2024/Conference — COLM_

### Official Review · Reviewer_qkT2 · 2024-05-10

**Rating:** 7
**Confidence:** 3
**Ethics Flag:** 1

**Summary:**

The authors feed existing labeled author verification datasets to ChatGPT to generate explanations for the labels, perform some sort of consistency checking between labels and explanations, then use LoRA to fine-tune LLMs on this generated data. Fine-tuned Llama 2 models outperform fine-tuned BERT and prompted GPT4 across 3 author verification datasets (IMDB, Twitter, Yelp). The addition of the explanation to the fine-tuning yields benefits across 3 LLMs (Llama1, OPT, Llama2). Generated explanations are evaluated both automatically and with human review (for a smaller sample).

Minor issues:
* There are many places where the citation style is wrong, e.g., "domains Halvani et al. (2019); Stamatatos (2016)" should be "domains (Halvani et al. 2019; Stamatatos 2016)". The "Author (Year)" form should only be used when the citation takes the place of a noun phrase.
* "A d key feature" => "A key feature"
* Define "PS+ prompting"
* "decisions. we have" => "decisions. We have"
* There's no need to spend space on the LoRA equation. Since you do not make any innovations over this formulation, just saying you're using LoRA for fine-tuning and citing the LoRA paper is enough.
* Tables are often far from where they are discussed. For example, Table 2 is first mentioned on page 8 but appears on page 5, and Table 3 is first mentioned on page 9 but appears on page 7.
* "InstructAV. we" => "InstructAV. We"
* I don't think Table 5 convincingly shows a correlation. Instead, show a scatter plot of explanation quality vs. classification accuracy for each dataset.

**Questions To Authors:**

* What is the background of the human evaluators? Were they blinded to which model's output they were evaluating?

**Reasons To Accept:**

* The work extends the chain-of-thought reasoning approach to author verification by introducing a new prompt that elicits 11 linguistic features in the answer explanations (since a simple "think step-by-step" wouldn't be appropriate)
* Asking LLMs to generate explanations of author verification predictions increases performance across 3 LLMs (Llama1, OPT, Llama2)
* The human evaluation shows that the explanations from the proposed fine-tuned model have better coverage of the requested linguistic features than prompted GPT-4.

**Reasons To Reject:**

* The IMDB, Twitter, and Yelp Reviews datasets are "widely used in AV studies" but the authors extract "11,000 samples from each dataset" instead of using a dataset split from prior work. In addition to the lack of direct comparability to prior work, I'm worried that the selected test set may be artificially easier, since the 11,000 samples are by definition the ones where ChatGPT could successfully explain the labels in the original dataset, as ones where there was inconsistency between ChatGPT's explanation and the original labels were discarded.
* The method used to check for consistency between labels and explanations is not described. There needs to be a formal description of this algorithm. As it is, I can only guess that it involves searching for some keywords like "similarities" or "differences", and cannot guess the extent of the keyword list nor what is done once those are found.
* Much of the first 3 pages reads like a sales brochure, describing the task being solved and talking up the approach, but not actually explaining what the technical approach is. The first real details about what exactly is being proposed don't appear until the 4th page.

---

> ### Author Rebuttal · Authors · 2024-05-31
>
> Thank you for your feedback and suggestions!
>
> **[W1]**
>
> For all 11,000 samples, ChatGPT could successfully explain the labels only when we explicitly provided the labels (“Text 1 and Text 2 are written by the same author/different authors”). However, PromptAV, another ChatGPT-based method, did not perform well in classification, indicating that without being given the label, ChatGPT cannot accurately classify or provide reliable explanations.
>
> To verify this, we conducted a supplementary experiment, randomly selecting another 1,000 samples from each original dataset without discarding any samples. We used the described models in the paper to perform classification tests on these samples. The results of these new tests are as follows:
>
> |Model|IMDB|Twitter|Yelp|
> |---|---|---|---|
> Classification
> Bert|0.703|0.660|0.629
> DistilBert|0.526|0.571|0.536
> AlBert|0.667|0.688|0.588
> InstructAV (LLaMA-1-7B)|0.621|0.603|0.539
> InstructAV (OPT-6.7B)|0.585|0.528|0.521
> InstructAV (LLaMA-2-7B)|0.913|0.753|0.691
> Classification & Explanation
> PromptAV-2shot (GPT-3.5)|0.611|0.656|0.625
> PromptAV-4shot (GPT-3.5)|0.612|0.654|0.578
> PromptAV-8shot (GPT-3.5)|0.528|0.638|0.583
> PromptAV (GPT-4-turbo)|0.715|0.727|0.625
> InstructAV (LLaMA-1-7B)|0.673|0.665|0.598
> InstructAV (OPT-6.7B)|0.735|0.701|0.633
> InstructAV (LLaMA-2-7B)|0.940|0.755|0.736
>
> From this supplemental experiment, we obtained similar results that InstructAV has better performance than all baseline models. The model's performance did not decline, so we believe that the removed samples did not truly impact our methodology.
>
> **[W2]**
>
> Our consistency check used keyword searches to verify authorship, restricting the model to follow a demonstrated output template (as in Table 7). We identified authorship from outputs by matching "written by the same author" (resp. “written by different authors”) with classification labels (Figure 1). All outputs were correctly aligned with these phrases. We will include a more detailed description in the updated paper.
>
> **[Q1]**
>
> All human evaluators hold at least a Bachelor's degree and are native English speakers. They were not informed about the origin of the explanations they assessed during the evaluation. The order of explanations in the questionnaire was randomized, with only the organizer knowing the model order.
>
> **[Minor issues]**
>
> Thank you for pointing out the grammatical mistakes in the paper. We will conduct a more thorough proofreading for the camera-ready version.

---

### Official Review · Reviewer_bxyx · 2024-05-10

**Rating:** 7
**Confidence:** 3
**Ethics Flag:** 1

**Summary:**

The paper presents a set of experiments on authorship verification using a newly developed framework based on large language models, new promising tools for various language processing tasks. The experiments are carried on using three well known corpora as benchmarks for authorship verification. Exploring the main purpose of large language models to generate text, the experiments included the generation of explanations of the model’s decision. Several stylometric characteristics had been used for text similarity evaluation. The experiments demonstrated high accuracy of author classification obtained by the proposed framework. The generated explanations have been evaluated automatically and manually to ensure their adequacy and consistency.

**Questions To Authors:**

Why do you use ChatGPT explanations as the ground truth?
Why the accuracy of classification & explanation is higher than for classification in Table 2?

**Reasons To Accept:**

The methodology of LLM prompting for author verification had been proposed in previous work but the current work developed this methodology, experimented on more datasets. The obtained results demonstrated the efficiency of the framework. The interesting part of the experiments is the evaluation of the quality of the generated explanations. Manual analysis is especially important; the quality of texts generated by LLM can vary considerably. Another advantage of the experiments was benchmark datasets used in the study; it allows adequate comparison with the related works.

**Reasons To Reject:**

Two types of experiments are combined in one paper; it could be clearer to make two papers with one type of experiments with fine-tuning LLMs and another one with prompt based classification with explanations. Combining various types of experiments the authors made the paper difficult to comprehend.
It is hard to say that automatic evaluation of explanations are effective.
The paper would be better with more detailed error analysis.

---

> ### Author Rebuttal · Authors · 2024-05-30
>
> Thank you for your feedback!
>
> **[Q1]**
>
> We found that none of the existing AV datasets include explanation labels. Collecting human annotations is unfeasible due to cost, time, and accuracy limitations. Given ChatGPT's capability to generate text analysis for multiple tasks, we decided to use it to generate explanations for our AV tasks.
> To ensure the quality of the explanations in our datasets, we employed a consistency verification method. We discarded "unreliable" explanations, where the authorship status was reported incorrectly (see our response to Q1 of Reviewer upBC for details). Only reliable explanations were kept as ground truth in our datasets.
> Furthermore, we asked human evaluators to assess the quality of these explanation labels. The results, presented in Table 4, demonstrate that the evaluators regarded these explanation labels as high-quality. Thus, we chose to use ChatGPT-generated explanations as the ground truth.
>
> Regarding the question about the incorporation of explanations, we posit that it can enhance the model's classification abilities. As discussed in the final subsection of 4.3, we observed that samples with higher quality explanations consistently achieved superior classification accuracy. Therefore, we believe that high-quality explanations positively impact the model's classification performance.

---

### Official Review · Reviewer_upBC · 2024-05-11

**Rating:** 7
**Confidence:** 5
**Ethics Flag:** 1

**Summary:**

## Summary
The paper proposes a new dataset that can be used for instruction fine-tuning for LLMs. As per my understanding, the process (briefly) goes as follows.
1) Collecting samples from three AV datasets.
2) Providing ChatGPT with a true label and ask it to generate explanation based on a specific template that includes 11 dimensions.
3) Verification of alignment between model answer about the similarity and the provided explanation. (**Not sure what happens with examples that are not-aligned).
4) Using the resulting dataset to fine-tune models (With LoRA)
5) Evaluation of the generated output (classification + explanation) using automatic metric and human evaluations.

### Quality
The authors investigate different aspects of evaluations for the performance. This includes both the quantitative and the qualitative evaluations.

### Clarity
There are a few details in the methodology that are not clear. Additionally, there are a few details that, IMO, would greatly enhance the readability of the paper:
1) Using the correct citation format across the paper. (Author's names should be inside the parentheses, except for cases when we are referring to the author's and not the work. E.g. _More details can be found in (Author et al 2020)_ **vs**. _Author (2020) discussed this_.
2) Giving more informative table captions.  Tables 2-5 all start with **performance of InstructAV on ...*. Instead, relevant captions such as Classification Accuracy, Token-overlap, ... etc.
3) In page 1, the last line: "*accuracy in authorship identification but ..*". This is probably a typo, authorship identification is a different task. Authors should avoid mixing up the two.
4) CoT and PS+ need citations.
5) Table~2: The caption mentions 2 datasets but the table shows 3.

### Originality
To me, the originality is in the instruction-tuning dataset. The use of this dataset results in an increase in the verification performance. Though, I have questions about the evaluation experiments.

### Significance
The field of AV would greatly benefit from new datasets that aim at understanding the problem. I think the presented dataset is helpful, but the evaluation details have a few unclear issues that require clarification. (Please see questions below)

P.S. The answer of the questions below is important to my review. I highly encourage the authors to answer the questions and not focus only on the preliminary decision.

# Scores have been updated after reading the responses

**Questions To Authors:**

. In Section 3.1, under consistency verification. I am not clear on this step: You enforce a template that tells the LM whether two text examples are for the same author or not, yet you mention that the model may start its response with the wrong classification label (hence, the verification step). What happens to such examples? are their labels corrected? or are they simply removed? If the template **prevents** the LM from generating the wrong response, then there would be no need for the verification step. Since there IS one, can you explain how are these examples handled?

. In Section 4.1 (Dataset), samples with incorrect linguistic analysis were removed. I think this is questionable. Why would such examples, which could be considered hard examples to distinguish be removed from the dataset? In fact, I think those are the examples that are more interesting. Selecting 10k examples that a language model has easily identified would make the task simpler and less realistic. Can you please clarify the intuition here?

. How sensitive is InstructAV to the choice of language model?

. Why is the results on IMDB very high compared to other datasets/or when other LMs are used?

. In table 4, ChatGPT results have the best Human evaluation results. Isn't this expected? Naturally, ChatGPT is a chat agent; instruction-tuned to have a dialogue and generate natural responses conditioned on the presented text. Why is this interesting to this study specifically?

**Reasons To Accept:**

1- Working towards explainability. IMO, it is very important to focus on the explanation of the AV results and not only present a performance number that is just higher than the previous SOTA.

2- The evaluation comprises both empirical results and qualitative analysis.

**Reasons To Reject:**

1- The analysis of the experiments is very limited and misses many interesting discussion points, e.g. the effect of the LM on the results.

1.1. In section 3.2, PromptAV and InstructAV are fine-tuned using different language models which brings up the effect of the LM itself on the results. This can be seen for InstructAV where it performs worse then BERT on two datasets when Llama-1 is used.

1.2.The second issue is that the performance on the IMDB dataset is very high when Llama-2 is used, but for the other datasets the results are close --and sometimes lower-- than the BERT baseline. It is actually troubling when the performance of the tool can only be seen with one specific language model and not across language models.

2. *Evaluation/Human Evaluation with ChatGPT*.

2.1. While using ChatGPT as a judge/metric has become very common, I don't see how we can consider its explanations as the ground-truth for this AV task. Consequently, using token-level metrics and BERTScore will only show the similarity of the responses among different language models but we all know that these metric have plenty of issues. Note that this is a *valid* experiment to conduct to evaluate the similarity between responses from language models. However, ChatGPT cannot be considered the gold-standard without human evaluation.

2.2. The human evaluation misses the main point. The human evaluation in this study is used to review the ChatGPT responses, but ChatGPT by design is able to generate answers that is of high quality. A more suitable human evaluation study would be to ask humans to generate explanation on why two texts are written by the same author and that would be the gold-standard and then we can compare it to ChatGPT responses. The problem is that such study would be very expensive to conduct and might have its own issues since humans do not do well on writing-style analysis tasks.

---

> ### Author Rebuttal · Authors · 2024-05-31
>
> Thank you for your thoughtful feedback and questions!
>
> **[Q1]**
>
> When ChatGPT was informed of the correct labels but failed to respond appropriately, we dropped such samples from our training set without correction. For instance, given the prompt "Text1 and Text2 are written by the same author. Please analyze…", ChatGPT might incorrectly analyze, “We find the following differences suggest they were written by different authors…", which is contrary to the truth. Such explanations, which we call “unreliable” explanations, were disregarded to enhance model robustness.
>
> **[Q2]**
>
> For training sets, as we explain in our response to Q1, only correctly classified examples were included to improve the models’ analytical abilities. We also used accurately classified examples for test sets to obtain reliable explanations and evaluate them. This approach was necessary due to the lack of suitable explanation labels in existing AV datasets and the impracticality of human annotation. We ensured uniform test sets across all models for fair performance comparisons.
> To address concerns about sample removal, we conducted supplemental experiments. Please refer to our response to W1 of Reviewer qkT2.
>
> **[Q3 & W1.1]**
>
> The performance of InstructAV directly correlates with the language models' inherent capabilities. e.g., LLaMA-2 always outperforms LLaMA-1.
>
> **[Q4 & W1.2]**
>
> We infer that InstructAV can better analyze long texts, like those in the IMDB dataset, which have more linguistic features than shorter texts.
>
> **[Q5]**
>
> Yes, this result was expected. We retained only reliable explanations in test sets after consistency checks. These were used to evaluate and compare InstructAV against ChatGPT. Both automatic and human assessments show InstructAV’s explanations are nearly as high-quality as reliable explanations.
>
> **[W2]**
>
> The intuition behind our approach is when ChatGPT can correctly determine the label, it can also provide a possible explanation. Therefore, we decided to use these explanations as explanatory labels in our dataset. We engaged human evaluators to rate these explanation labels to validate this intuition. The outcomes of this evaluation, detailed in Table 4 under 'Ground Truth,' indicate that these explanation labels are consistently recognized as high-quality by the evaluators. We used the BERTScore metric to test whether the explanations generated by InstructAV are semantically consistent with these explanation labels.

---

> > ### Author Response · Authors · 2024-06-05
> >
> > Dear Reviewer upBC,
> >
> > We hope that you've had a chance to read our response. We would really appreciate a reply to let us know if our response and clarifications have addressed the issues raised in your review, or if there is anything else we can address.
> >
> > Sincerely,
> >
> > Authors

---

> > > ### Comment · Reviewer_upBC · 2024-06-05
> > > **Brief summary**
> > >
> > > Dear authors,
> > >
> > > Thank you for the response. I did read your reply to my comments and to the other reviewers' as well. I believe that the experiment on 1000 examples (response qkT2 below) is very important AND is the correct setup to conduct the experiments. It also shows that with this setup similar observations about the performance of baselines and proposed methods still hold.
> > >
> > > Additionally, I think that most of the issues can be solved either by rewording and making careful claims (specifically about human evaluation being the ground truth), and some doable reorganization of the experimental sections (if you choose to do so).
> > >
> > > Finally, please consider the comments that I've provided about clarity above.

---

> > > > ### Author Response · Authors · 2024-06-05
> > > >
> > > > Thank you for your detailed comments.
> > > >
> > > > **[Q1]**
> > > >
> > > > Yes, we will include these details in the updated paper.
> > > >
> > > > **[Q2]**
> > > >
> > > > Thank you for your suggestions. In the updated paper, we will adopt your recommendation to divide the classification and Classification+Explanation sections into two parts. For the classification section, we will use the new test set without discarding any samples. For the Classification+Explanation section, we will use the previously selected samples and provide additional explanations.
> > > >
> > > > **[Q3 & W1.1]**
> > > >
> > > > The performance of InstructAV fine-tuning is correlated with the capabilities of the base model. These capabilities are influenced by factors such as the model size, the size and quality of the pretraining corpus. For instance, we employed 7B models for both LLaMA-2 and LLaMA-1, but LLaMA-2 performed better due to being trained on more tokens and having longer context lengths. Therefore, if we choose a base model with a larger size and a more extensive pretraining corpus, we can expect better performance.
> > > >
> > > > **[Q4 & W1.2]**
> > > >
> > > > Thank you for your insightful comment. I appreciate your perspective and the opportunity to clarify our approach. We understand your point that computational approaches often outperform humans in authorship analysis tasks, as evidenced by works such as Stamatatos (1999). Our intent in using human evaluation was indeed to perform a high-level check to ensure that the explanations generated by ChatGPT were coherent and aligned with the content of the text, rather than to serve as a comprehensive evaluation of authorship features. Specifically, our primary goal was to verify that these explanations were sensible and relevant to the texts being analyzed.
> > > > We acknowledge that frequency and other nuanced features play a crucial role in authorship analysis, which computational methods are better suited to capture. As per your earlier comments, a more suitable human evaluation study would involve asking humans to generate explanations on why two texts are written by the same author, which could then serve as a gold standard for comparison with ChatGPT's responses. However, such a study would be very expensive to conduct and might have its own issues, given that humans do not perform well on writing-style analysis tasks.
> > > > Given these limitations and constraints, we considered ChatGPT-generated explanations to be a reasonable and cost-effective alternative; minimally, we have verified that the explanations generated by ChatGPT were coherent and aligned with the text content.
> > > >
> > > > In our updated paper, we will make these distinctions clearer and provide more context around the purpose and scope of our human evaluations. We will also highlight the limitations of our approach and elaborate on the role of computational methods in capturing the finer details of authorship features.
> > > >
> > > > **[Brief summary]**
> > > >
> > > > Once again, thank you for the valuable questions, which have greatly refined our paper. We will revise our paper to address your comments and questions, including those about clarity, to improve overall readability.

---

> > ### Comment · Reviewer_upBC · 2024-06-05
> > **Detailed comment**
> >
> > Thank you for the detailed response.
> >
> > Q1) I see. Would be great if you can clarify this in the paper as well.
> >
> > Q2) Thank you for the clarification. I see an issue here where certain samples were dropped because chatGPT did not do well on them. To me, this is what happened: "we have a test set and some examples are problematic to ChatGPT so we are going to remove them and consider the remaining samples as our new test set". Perhaps, this would work if all the models are of the same family, but comparing with the baselines (which are very competitive even after subsampling) would not be fair.
> >
> > You address this point in "response to W1 of Reviewer qkT2" which is very important and in fact, I think that's how your main experiment should be. This should make the separation between classification (the typical verification task) experiment and explanation experiment much clearer. In the classification experiments below, the baseline BERT outperforms the other methods including yours except when Llama-2 is used. Conditioning the effectiveness of the proposed approach on the use of one specific language model is problematic.
> >
> > IMO, a much stronger case for your paper would be either:
> > 1) propose InstructAV as a data collection method where you show that using the selected samples, fine-tuned models will result in better performance. This way, you are not claiming a new method and therefore don't have to explain why one language model performs better than the other.  Or,
> > 2) Have a stronger separation between classification only and clarification with explanation. I would suggest splitting the experiments (and table 2) into two parts: Classification where you show the performance on ALL the test samples like you did in the response below, and Classification+Explanation on the selected samples where you compare only with methods that provide explanation. In step two, you can justify the drop in performance when other language models by offering an explanation to the results. In Authorship Analysis, it is a well-known phenomena that explainable methods have lower performance and SOTA methods are harder to explain (especially to the layman person).
> >
> > Q3 & W1.1) Thank you. This is clear from the results, but why? is it because of the model size? or the characteristics of the model? If I want to use your method with another model, how do I choose that model? just the biggest model available? I am interested in more than describing what the table shows.
> >
> > [Q4 & W1.2] That's a strong claim. You cannot infer that without sharing descriptive stats of the data, e.g. avg length in words for example.
> >
> > [W2: ChatGPT can correctly determine the label, it can also provide a possible explanation.] I respectfully disagree. Please let me clarify that for authorship analysis tasks "humans" do not do well compared to computational approaches. (see Stamatatos  99, for example). Therefore, when you generate an explanation such as "uses passive voice", or "passionate about the topic" or some generic explanation, humans would tend to agree unless such note is completely wrong. What matters more in authorship analysis is the frequency of such features. What I was trying to say that, human evaluation in your case is a high-level check that whatever ChatGPT generated is actually in the text but not more than that. It is very different from the typical "human evaluation" for other tasks. Even BERTScore, it is just showing that similar models generated similar text and that on its own has its problems.
> >
> > Stamatatos, E., Fakotakis, N., & Kokkinakis, G. (1999, June). Automatic authorship attribution. In Ninth conference of the European Chapter of the Association for Computational Linguistics (pp. 158-164)

---

### Official Review · Reviewer_GRur · 2024-05-18

**Rating:** 5
**Confidence:** 4
**Ethics Flag:** 1

**Summary:**

This paper introduces InstructAV, a novel approach for authorship verification (AV) tasks using Large Language Models (LLMs) with parameter-efficient fine-tuning (PEFT). InstructAV aims to improve both accuracy and explainability in determining whether two given texts are written by the same author. The framework consists of three main steps: data collection, consistency verification, and LLM fine-tuning using the LoRA method. The authors curate three instruction-tuning datasets with reliable linguistic explanations and demonstrate the effectiveness of InstructAV through automated and human evaluations.

**Questions To Authors:**

How do the authors address the potential data contamination problem?

The potential risk of data leakage is not discussed. The data might have been included in the LLMs' training corpus. I wonder if there are some tests showing the data memorization possibilities and how the authors address the potential data contamination problem.

**Reasons To Accept:**

Pros:
1. The InstructAV framework significantly improves AV classification accuracy compared to baseline models, particularly when paired with LLaMA-2-7B.
2. The approach generates high-quality explanations for AV predictions, providing valuable insights into the model's decision-making process.
3. The consistency verification step ensures alignment between classification labels and linguistic explanations, enhancing the reliability of the generated explanations.
4. InstructAV demonstrates state-of-the-art performance on AV tasks across various datasets, offering both high classification accuracy and enhanced explanation quality.
5. The curated instruction-tuning datasets with reliable linguistic explanations are valuable resources for advancing research in the AV field.

**Reasons To Reject:**

Cons:
1. Table 4 hard to understand. It is strongly suggested that the authors should explain the evaluation metrics for “Coverage Relevance Reasonableness Persuasiveness” clearly in the table. Otherwise, it is hard for readers to understand.
2. The current baselines are old. It is strongly suggested that the authors include LLMs as baselines considering LLMs have achieve a good performance in Authorship Attribution[2,3]. Otherwise, it is hard to evaluate the effectiveness of the proposed method.
3. No code is provided. It is strongly encouraged that the authors release the code for reproduction.
4. Not sure whether the authors run the baselines multiple times to ensure the convincingness. If the authors run multiple times, it is suggested the authors provide more details.
5. It is encouraged the authors conduct statistical testing for the experiments.
6. Related works are incomplete [1,3,4,5]. It is strongly suggested that the authors conduct a comprehensive literature review.

[1] VeriDark: A Large-Scale Benchmark for Authorship Verification on the Dark Web https://arxiv.org/abs/2207.03477

[2] Who Wrote it and Why? Prompting Large-Language Models for Authorship Verification https://arxiv.org/abs/2310.08123

[3] Can Large Language Models Identify Authorship? https://arxiv.org/abs/2403.08213

[4] Can Authorship Representation Learning Capture Stylistic Features? https://arxiv.org/abs/2308.11490

[5] Contrastive Disentanglement for Authorship Attribution https://dl.acm.org/doi/10.1145/3589335.3652501

---

> ### Author Rebuttal · Authors · 2024-05-31
>
> Thank you for your helpful feedback and questions.
>
> **[Q1]**
>
> Section 4.2 details "Coverage, Relevance, Reasonableness, Persuasiveness."  To enhance clarity, we will include a summary of these metrics in the caption of Table 4.
>
> **[Q2]**
>
> Our baseline "PromptAV" is an LLM-based model referenced in [2]. We did not initially include the model in Huang et al. [3] since it was published after our initial experiments, but have now incorporated it. Utilizing the LIP prompting technique from [3], we performed additional tests:
>
> |BaselineModel|IMDB|twitter|yelp|
> |----|----|----|----|
> GPT-4Turbo+LIPprompt|0.739|0.616|0.633
> LLaMA-2-70B+LIPprompt|0.534|0.559|0.524
> Mistral+LIPprompt|0.504|0.541|0.526
> Ours
> InstructAV(LLaMA-2-7B)|0.938|0.747|0.661
>
> InstructAV still outperforms all of these baselines. We will add these new results to our paper.
>
>
> **[Q3]**
>
> We will include a GitHub link to our code and all datasets in the camera-ready version.
>
> **[Q4]**
>
> To ensure our results' robustness, we repeated each experiment three times, calculating the standard deviation (SD) for each model. Below is a table showing average accuracies and SD (in brackets):
>
> |Model|imdb|twitter|yelp|
> |----|----|----|----|
> Bert|0.677 (0.0124)|0.702 (0.0021)|0.622 (0.0020)
> Distilbert|0.526 (0.0125)|0.575 (0.0065)|0.543 (0.0036)
> Albert|0.642 (0.0040)|0.701 (0.0023)|0.601 (0.0020)
> PromptAV-2shot (GPT-3.5)|0.623 (0.0397)|0.628 (0.0147)|0.534 (0.0064)
> PromptAV-4shot (GPT-3.5)|0.635 (0.0265)|0.667 (0.0163)|0.544 (0.0080)
> PromptAV-8shot (GPT-3.5)|0.601 (0.0070)|0.648 (0.0075)|0.564 (0.0081)
> PromptAV (GPT-4)|0.755 (0.0075)|0.729 (0.0070)|0.597 (0.0065)
> InstructAV (LLAMA-1-7B)|0.825 (0.0289)|0.625 (0.0065)|0.596 (0.0104)
> InstructAV (OPT-6.7B)|0.744 (0.0095)|0.714 (0.0070)|0.575 (0.0140)
> InstructAV (LLAMA-2-7B)|0.937 (0.0017)|0.745 (0.0063)|0.693 (0.0442)
>
> The small SD values across all datasets indicate consistent experimental results.
>
> **[Q5]**
>
> We tested statistical significance between the top-performing baseline model, PromptAV-2shot-GPT4, and our's InstructAV (LLaMA2-7B):
>
> ||IMDB|Twitter|Yelp
> |----|----|----|----|
> p-value|4.630e-8|1.597e-4|1.516e-4
>
> All p-values are <0.05. This confirms significant improvements in our InstructAV model.
>
> **[Q6]**
>
> We'll add the suggested references and conduct a detailed literature review in our updated paper.

---

> ### Comment · Reviewer_GRur · 2024-06-03
> **How do the authors address the potential data contamination problem?**
>
> The potential risk of data leakage is not discussed. The data might have been included in the LLMs' training corpus. I wonder if there are some tests showing the data memorization possibilities and how the authors address the potential data contamination problem.

---

> > ### Author Response · Authors · 2024-06-05
> >
> > Thank you for your reply and comment.
> >
> > To investigate potential data contamination, we conducted 0-shot, 2-shot, 4-shot, and 8-shot tests on each dataset using the original, un-tuned LLaMA-2-7B, LLaMA-1-7B, and OPT-6.7B models. Each experiment was performed three times, and the average results are presented below:
> >
> > | |Model | IMDB | Twitter | Yelp |
> > |----|----|----|----|----|
> > 0-shot |LLaMA-1-7B | 0.000 | 0.000 | 0.000
> > | | OPT-6.7B | 0.000 | 0.000 | 0.000
> > | |LLaMA-2-7B | 0.006 | 0.010 | 0.021
> > 2-shot |LLaMA-1-7B | 0.189 | 0.097 | 0.226
> > | | OPT-6.7B | 0.258 | 0.179 | 0.297
> > | |LLaMA-2-7B | 0.309 | 0.336 | 0.397
> > 4-shot  |LLaMA-1-7B | 0.014 | 0.020 | 0.127
> > | | OPT-6.7B | 0.191 | 0.220 | 0.217
> > | |LLaMA-2-7B | 0.279 | 0.350 | 0.375
> > 8-shot |LLaMA-1-7B | 0.003 | 0.007 | 0.002
> > | | OPT-6.7B | 0.005 | 0.008 | 0.003
> > | |LLaMA-2-7B | 0.020 | 0.033 | 0.025
> >
> > The results indicate that the original models struggle to perform AV tests effectively on our dataset, especially in the 0-shot setting where the outputs are highly random. Both LLaMA-1-7B and OPT-6.7B failed to make correct judgments, and LLaMA-2-7B showed weak judgment capabilities. The best performance was observed with the 2-shot models, which could produce outputs following the template format, "The correct answer is yes/no...[Analysis]." However, even when the format was correct, the accuracy was still very low, with the best result (LLaMA-2 on the Yelp dataset) being lower than a random guess. Consequently, we postulate that it is unlikely that our dataset was included in the training data of the original models.

---

> > > ### Author Response · Authors · 2024-06-06
> > >
> > > Dear Reviewer GRur,
> > >
> > > We hope that you've had a chance to read our comment. We would really appreciate a reply to let us know if our response and clarifications have addressed the issues raised in your previous comment, or if there is anything else we can address.
> > >
> > > Sincerely,
> > >
> > > Authors

---

### Author Response · Authors · 2024-06-07

Thank you to all the reviewers for your time and effort in reviewing our paper. Your suggestions are very useful for improving the quality of our paper. Many thanks!

---

### Decision · Program_Chairs · 2024-07-10

**Decision:**

Accept

**Comment:**

The authors propose an approach for authorship verification tasks using Large Language Models with parameter-efficient fine-tuning. The approach is based on three main components: data collection, consistency verification, subspace adaptation (LORA). The proposed approach allows the authors to obtain higher author verification classification accuracy on benchmarks, as well as high-quality explanations of the predictions. The reviewers praised the impressive results, yet brought up some concerns. The authors submitted a response that addressed the concerns, including additional experimental results, and detailed on-point answers. The reviewers are overall enthusiastic about the paper, and the paper can be considered for a spotlight presentation or an oral presentation.

Clear Accept.